# Endoscopic Submucosal Dissection (ESD) of Upper Gastrointestinal Carcinomas: An Integrated Clinical and Pathological Perspective

**DOI:** 10.3390/jcm14248817

**Published:** 2025-12-12

**Authors:** Alexander Ziachehabi, Maximilian Worm, Drolaiz H. W. Liu, Philipp Pimingstorfer, Rupert Langer

**Affiliations:** 1Department for Internal Medicine 4, Ordensklinikum Linz-Barmherzige Schwestern, 4020 Linz, Austria; alexander.ziachehabi@ordensklinikum.at; 2Department of Pathology and Molecular Pathology, Johannes Kepler University Linz, 4021 Linz, Austria; maximilian.worm@gmx.at (M.W.); drolaiz.liu@kepleruniklinikum.at (D.H.W.L.); 3Kepler University Hospital GmbH, 4021 Linz, Austria; philipp.pimingstorfer@kepleruniklinikum.at; 4Department of Pathology, GROW Research Institute for Oncology and Reproduction, Maastricht University Medical Center, 6229 HX Maastricht, The Netherlands; 5University Clinic for Internal Medicine 2, Johannes Kepler University Linz, 4040 Linz, Austria

**Keywords:** endoscopic resection, endoscopic submucosal dissection, esophagus, stomach, pathology

## Abstract

Endoscopic submucosal dissection (ESD) has revolutionized the management of early upper gastrointestinal (GI) carcinomas. While technically demanding, it offers, in experienced hands, definitive local therapy for early GI neoplasia by allowing complete En bloc resection of mucosal and superficially invasive neoplasms, thus enabling precise histopathological risk stratification and organ preservation. Appropriate patient selection relies on meticulous endoscopic assessment using high-definition and image-enhanced endoscopy to define lesion boundaries and predict invasion depth. The principal indications include high-grade intraepithelial neoplasia and early carcinomas without endoscopic evidence of deep submucosal invasion or lymph node metastasis risk factors. Pathological analysis of the resection specimens includes histological typing and grading per WHO classification and precise assessment of invasion depth—in case of submucosal invasion measurement in micrometers—and evaluation of margin status and lymphovascular invasion. The presence of risk factors such as deep invasion in the submucosa, poor differentiation, or lymphovascular invasion may require additional surgery, guided by validated risk scores such as the eCura system. This narrative review summarizes current clinical and pathological practices for ESD in upper GI lesions. This includes the discussion of technical and biological challenges and the need of accurate assessment of risk factors for systemic metastatic spread and local recurrence as a limitation for this sophisticated but highly effective therapeutic method.

## 1. Introduction

Early gastroesophageal carcinomas, including superficial esophageal squamous cell carcinomas (ESCCs), esophageal adenocarcinomas (EACs), adenocarcinomas of the gastroesophageal junction (GEJ), and gastric cancers (GCs), are intramucosal carcinomas or superficially invasive carcinomas without or with only minor risk for lymph node metastases [1,2]. Historically, esophagectomy or subtotal gastrectomy was the standard of care for these early cancers [3,4,5]. However, although curative for most patients, esophagectomy is associated with significant morbidity and mortality and low quality of life postoperatively. The development of endoscopic resection (ER) techniques, with endoscopic mucosal resection (EMR) as first method described and applied in Japan during the 1980s, established local excision as an effective therapy for superficial gastric and esophageal cancers, later also expanding to other GI lesions [6,7,8,9,10]. Despite its success, EMR presented limitations for larger or more complex lesions, prompting the evolution of endoscopic submucosal dissection (ESD) in Japan in the following decades [7]. ESD enabled true En-bloc resection using electrosurgical knives, improving local control and pathological assessment for lesions not suitable for piecemeal EMR. Today, both EMR and ESD are integral tools in the endoscopic portfolio for the treatment of early upper gastrointestinal (GI) carcinomas. Continuous technical improvements and expansion of their indications worldwide has helped to significantly increase the quality of life of patients with localized gastroesophageal cancers who did not need to undergo surgery for an oncologically satisfactory treatment [6,7,11,12,13,14,15].

Integration of clinical and pathological perspectives is essential for optimal patient care. Clinically, the identification of appropriate indications relies on careful endoscopic assessment and patient factors. From a pathological point of view, the meticulous examination of ESD specimens provides not only accurate histopathologic classification of the lesions but also precise evaluation of invasion depth, lymphovascular invasion, and margin status for malignant lesions [16,17,18]. These histological risk factors predict the two main outcomes: risk of lymph node metastasis and risk of residual disease at the ER site.

This article aims to provide a comprehensive overview of ESD for carcinomas of the upper gastrointestinal tract by synthesizing current clinical practices and pathological evaluation and reporting standards. It will also address limitations and challenges of histopathologic work up and finally highlight the importance of a bilateral dialog between clinics and pathology.

## 2. Methods

This article was designed as a narrative review with the aim of synthesizing current practice of the application of ER techniques, particularly ESD on upper GI carcinomas, and pathology work up and reporting of the respective resection specimens.

A comprehensive, non-systematic search of the literature was conducted across major databases, including PubMed, covering publications from 2019 to 2025. Relevant studies, reviews, and meta-analyses were identified using combinations of the following keywords: “ESD”, “EMR”, “esophagus/esophageal”, “gastric”, “cancer”, “histopathology”, “risk”, “Barrett”, “endoscopic/endoscopy”, and “guidelines”. Articles were then selected based on their relevance to one or more of the following themes: application of ER techniques, particularly ESD; histopathology work up and reporting; and guidelines. Due to the character of a narrative review, no formal quality assessment or meta-analysis was performed. Most of the references finally included are reviews, meta-analyses, and guidelines.

## 3. Histopathology of Lesions Suitable for Endoscopic Resections

The histologic spectrum of lesions amenable to ESD or ER in general comprises primarily benign lesions and both precancerous conditions and early-stage malignancies. In the esophagus, the principal histologic entities are squamous and glandular lesions [19,20,21].

Squamous intraepithelial neoplasia, graded as low- or high-grade dysplasia, represents the precursor stage to squamous cell carcinoma, which itself may be encountered in early, intramucosal form suitable for ER or in carcinomas with superficial invasion of the submucosa without risk factors for lymphatic metastases, as discussed more in detail later in this paper [22].

In the glandular compartment, Barrett’s esophagus gives rise to a sequence of dysplastic changes, ranging from indefinite for dysplasia through low-grade and high-grade dysplasia, ultimately progressing to adenocarcinoma [22]. In the stomach, the lesions relevant to ER include a variety of adenomatous dysplastic proliferations. Intestinal-type adenomas are the most frequent, besides foveolar-type and pyloric gland adenomas. Dysplasia may also present outside a polypoid adenoma, as flat or depressed lesions. Gastric carcinoma is subtyped according to the WHO classification into tubular, papillary, poorly cohesive (including signet-ring cell carcinoma), and mucinous carcinomas and several special subtypes [21,23,24]. In addition to the WHO classification, the Laurén classification (intestinal vs. diffuse type) [25] and the Japanese classification [26] with similar detailed subtypes are widely accepted. In addition, small, well-differentiated neuroendocrine tumors (NETs) of grade 1 or 2 [27] may be resected endoscopically, as may hyperplastic or hamartomatous polyps if they harbor focal dysplasia, or other benign, mesenchymal lesions arising in the mucosa [28]. This review, however, primarily focusses on high-grade dysplastic lesions and squamous cell carcinomas and adenocarcinomas.

## 4. Clinical Indications

Endoscopic resection has emerged as the preferred first-line approach for appropriately selected patients with (high-grade) dysplastic lesions or early carcinomas across international guidelines—including those from the European Society for Medical Oncology (ESMO) [29,30], the National Comprehensive Cancer Network (NCCN) [31,32], and the Japanese Gastric Cancer and Esophageal Cancer guidelines [33,34,35]. Surgical resection is reserved for carcinomas with higher risk of insufficient oncological outcomes, e.g., disseminating disease or unresectable lesions by endoscopic means. This stratification reflects a balance between cure rates, functional preservation, and procedural safety.

Apart from local indications such as multifocality, for lesions in locations or with conditions limiting complete endoscopic clearance, the risk of lymphatic spread with subsequent node metastases is the major determinator for choosing primary surgery as a curative therapeutic option. Lymphatic spread, in turn, is associated with aggressive tumor features (poor differentiation), larger tumor size, and deeper tumor infiltration, which should be determined before endoscopic intervention [36,37,38]. The final assessment of risk factors with a subsequent definitive therapeutic decision (i.e., follow-up for oncologically sufficient and definite curative excision vs. completing surgery for situations with high-risk of failure of local tumor control, of lymphatic spread, or recurrence) is then performed on the resection specimen by histopathology [16].

## 5. Preintervention Assessment

The most important part of the pre-intervention assessment is meticulous endoscopic evaluation. This is typically performed with high-definition white-light endoscopy to clearly delineate the lesions and their boundaries [11,39]. Image-enhanced modalities, including narrow-band imaging (NBI), blue laser imaging, or chromoendoscopy, further enhance visualization of subtle mucosal and vascular details. Detection of irregular microvascular (vascular) or micro-surface (epithelial) patterns can non-invasively predict which lesions are likely to harbor high-grade dysplasia, intramucosal carcinoma, or invasion into the submucosa with high accuracy [40]. Particularly, the presence of a demarcation line separating the abnormal area from normal mucosa in combination with irregular microvascular or micro-surface patterns is highly predictive of neoplasia. Normal mucosa exhibits regular, well-organized microvascular and micro-surface features. In high-grade dysplasia or early cancer, the microvasculature becomes irregular, dilated, and tortuous (“corkscrew” vessels), or shows mesh-like, branching patterns, while the epithelial surface loses its uniformity and regular pit pattern, showing chaotic or disrupted features [41,42]. Specific features linked with submucosal invasion include non-dense vessel density, negative vessel regularity, and increased vessel meandering [43]. Visible tumor lesions identified during endoscopic assessment are described according to the Paris classification, which categorizes them as polypoid (type I), flat (type II), or excavated (type III). The flat, type II lesions can be further subdivided: those with slight elevation less than 1.3 mm (IIa), entirely flat (IIb), or superficially depressed (IIc). Flat, depressed, and excavated lesion types are associated with a considerably increased probability of submucosal invasion, which impacts the suitability for endoscopic resection [44].

Endoscopic ultrasound (EUS) is not always required routinely, but in certain cases it serves as an important adjunct, particularly for esophageal lesions [45,46]. Here, EUS can help to additionally assess the depth of tumor infiltration and exclude frank involvement of the muscularis propria, However, EUS accuracy drops for subtle submucosal invasion and is not recommended routinely for further assessment of these lesions. Routine cross-sectional imaging (CT, MRI, and PET-CT) is not typically indicated before endoscopic resection, except in scenarios with suspected advanced disease [29,30].

For histologic confirmation, targeted samples from representative lesion areas are preferable. Taking multiple or deep biopsies may risk inducing fibrosis and complicating subsequent ESD. This is in contrast to the recommendations for advanced carcinomas, where multiple biopsies are required for subsequent biomarker analysis, particularly in view of the intratumoral heterogeneity in these larger tumors [47,48]. In histological biopsy diagnosis, a distinction is made between squamous and glandular lesions, the presence or grade of dysplasia, and the presence of invasive carcinomas [21]. For glandular lesions, reporting according to the Vienna Classification is recommended, which recognizes the following: Category 1: Negative for neoplasia/dysplasia; Category 2: Indefinite for neoplasia/dysplasia (i.e., uncertain diagnosis, often due to inflammation or regeneration obscuring clear assessment, with close follow-up monitoring recommended); Category 3: Non-invasive low-grade dysplasia; Category 4: Non-invasive high-grade dysplasia, carcinoma in situ, intramucosal carcinoma, or suspicion of invasive carcinoma; Category 5: Invasive neoplasia (intramucosal or submucosal infiltration, or beyond) [49] (Table 1).

EMR may be sufficient for non-ulcerated, polypoid, well- or moderately differentiated smaller lesions < 15–20 mm without endoscopic features of deep invasion. ESD is favored for smaller, superficial lesions—but usually less than 2 cm in the esophagus or less than 2–3 cm in the stomach—that are well- or moderately differentiated by histology and lack ulceration or a flat or depressed macroscopic morphology [50]. These features correlate with a high likelihood of curative resection and a negligible risk of lymph node metastasis. Other high-risk histology features beyond size, subtype of the lesion, or differentiation, namely lymphovascular invasion or deep submucosal penetration—as described more in detail below—can be assessed by histology only after ER has already been performed [16].

The location of the lesion may also contribute to therapeutic decision-making: ideal for ESD are better accessible and technically favorable areas such as the gastric antrum, mid/distal esophagus, or lesser curvature of the stomach. These sites facilitate easier scope maneuverability, En bloc resection, and lower complication rates. In contrast, the gastric cardia, fundus, greater curvature, upper/middle esophagus, or near anatomic bends and previous surgical sites are anatomically challenging regions posing significant technical difficulties for ESD. These locations increase the risk of incomplete resection, perforation, and procedural failure. In such cases, surgery may be preferred, especially if expertise or specialized equipment is limited. Algorithms for clinical decision-making are presented in Figure 1.

## 6. Endoscopic Submucosal Dissection Procedure

ESD is performed under high-definition endoscopic visualization [11,51]. The procedure begins with careful characterization and delineation of the lesion using advanced imaging modalities, including white-light endoscopy, chromoendoscopy, or narrow-band imaging, to define margins with high accuracy. After demarcating the lesion’s periphery, a submucosal injection—typically consisting of a viscous or saline-based solution mixed with epinephrine and a contrast dye—is administered. This creates a safety cushion that elevates the lesion from the muscularis propria, thereby reducing the risk of deep thermal injury and potential perforation [10,52] (Figure 2A,B).

A circumferential mucosal incision is then created along the marked margins using an electrosurgical knife. Subsequent careful submucosal dissection is performed to separate the lesion En bloc from the underlying tissue, proceeding along the relatively avascular plane to preserve adjacent healthy mucosa. During dissection, visible vessels are proactively coagulated with hemostatic forceps to minimize intraprocedural bleeding, and continuous irrigation as well as transparent distal attachments (caps) are employed to optimize visualization of the dissection plane [12,15,53,54] (Figure 2C–F).

Potential complications include bleeding, perforation, and stricture formation, which are generally infrequent but require expert management when encountered [51].

A sign for deeper tumor infiltration is the lack of lifting after submucosal injection. Apart from tumor intrinsic changes caused by tumor infiltration and associated stromal desmoplasia, fibrosis, due to prior biopsies, previous endoscopic resections, or chronic inflammation, can make the submucosal layer tough and less distinct. Marked fibrosis causes poor lesion lifting upon submucosal injection, makes dissection more complex, and heightens the risk of perforation or incomplete removal. In cases of severe fibrosis, ER can become technically unfeasible or unsafe, requiring longer procedure times and increasing adverse event rates [52].

Once the lesion is resected, the specimen must be immediately retrieved and properly prepared for later histopathology work up (Figure 2D,E). Ideally the specimen should be pinned, mucosal side up, on a flat board usually made of cork, Styrofoam, or silicone. This step preserves the anatomical orientation, prevents curling, and allows for reproducible spatial correlation between endoscopic and histological findings. Specimen shrinkage can make margin assessment difficult if not stretched properly before fixation. After pinning, the specimen is immersed in 10% neutral-buffered formalin, typically for a duration of 24 to 48 h. A fixative volume of at least 10 times the specimen volume is recommended. Fixation delay or uneven penetration can result in artifacts such as epithelial detachment, obscuring diagnostic features [22].

## 7. Macroscopic Work up in Pathology

Upon receipt in the pathology laboratory, the specimen is photographed, ideally with a ruler for scale and with annotations to indicate the lesion and margins (Figure 3). Macroscopic assessment includes documentation of lesion size, ulceration, depth, and proximity to margins if possible. Marking dyes (e.g., India ink or tissue-marking dyes in different colors) are applied to lateral and deep margins, for accurate determination of margins by subsequent histology work up.

The specimen is then serially sectioned perpendicular to the long axis at intervals of 2 to 3 mm. Careful placement and orientation of the sections in the cassettes is of utmost importance for the correct histopathology assessment. In most cases standard tissue cassettes can be used for further histology processing including paraffin embedding and sectioning. For large specimens, the use of macro-cassettes or L-shaped embedding molds may help to optimize tissue handling and reduce the number of blocks, while preserving orientation and diagnostic yield.

## 8. Microscopic Evaluation

### 8.1. Work up

Microscopic work up is based on perpendicular sections through the lesion and margins [55] (Figure 3A–D). The whole specimen should be histologically analyzed. Usually, standard Hematoxylin-Eosin (HE) staining is sufficient to provide all necessary information for the diagnosis of the lesions and the assessment of risk factors. In selected cases, immunohistochemistry for cytokeratin, particularly AE1/AE3, may be useful for the detection of single infiltrating cells and for demonstration of a subtle infiltrating poorly cohesive carcinoma (particularly in the stomach) or tumor buds. It can be helpful in delineating the extent of the carcinoma and demonstrating submucosal invasion where it is subtle or obscured by inflammatory cell infiltrates. Spindle cell (squamous) carcinoma may express cytokeratin, aiding distinction from primary sarcomas and spindle cell melanoma. In the case of poorly differentiated carcinoma, high-molecular-weight cytokeratin (e.g., CK5/6), p63, and/or p40 (which are all typically positive in squamous cell carcinoma) may help differentiate squamous cell carcinoma from adenocarcinoma and basaloid squamous carcinoma of the esophagus from the rare adenoid cystic carcinoma. Simple mucin stains (alcian blue, periodic acid–Schiff, or mucicarmine) may aid in the differentiation of adenocarcinoma from poorly differentiated squamous cell carcinomas. For precancerous lesions, overexpression of p53 and a high Ki-67 labeling index in atypical cells beyond the basal layers of the epithelium for squamous lesions or at the surface for glandular lesions suggest neoplastic progression. This can help resolve diagnostic uncertainty in areas of inflammation or regenerative atypia [56,57]. Immunohistochemistry for smooth muscle markers (desmin or another smooth muscle marker) demonstrates the smooth muscle of vessel walls when venous invasion is suspected. It is also useful for highlighting and delineating the muscularis mucosae in areas with suspected submucosal invasion (particularly in the setting of duplicated muscularis mucosae in Barrett’s esophagus). Alternatively, histochemical stains may be used for highlighting the muscularis mucosae (trichrome and Elastica van Gieson). Vascular markers can be helpful in detecting lymphovascular vessel invasion and may demonstrate this feature when it is not seen on HE-stained sections (e.g., D2-40, CD34, CD31, and ERG) [55].

### 8.2. Tumor Diagnosis

Tumor diagnosis and typing should be performed according to the WHO classification [23]. Other classification systems such as the Laurén classification for gastric cancer and the Japanese classifications for esophageal and gastric cancers do exist and may be applied as well [21]. Grading is also performed according to the respective guidelines. In the western world, poor differentiation or aggressive histology, such as poorly cohesive or signet-ring cell components may occur only rarely in ER specimens, since they would represent contraindications for local excision if diagnosed before the intervention, in contrast to Asia. Other potentially adverse prognostic morphologic tumor features such as tumor budding may be reported. However, due to lack of data and the observation that tumor budding is rather a phenomenon of advanced carcinomas, reporting is not generally recommended in contrast to the lower gastrointestinal tract.

### 8.3. Assessment of Invasion Depth

In ER specimens, invasion depth is usually limited to the mucosa or submucosa. For intramucosal carcinomas, two systematic methods of differentiating depth of invasion into the different layers of the mucosa have been introduced (Figure 4). The first method recognizes three levels (M1–M3) [58]. M1: carcinoma confined to the intraepithelial layer, without invasion into the underlying lamina propria mucosae; M2: invasion into the lamina propria mucosae, but not reaching the muscularis mucosae; M3: infiltration into or through the muscularis mucosae, but not into the submucosa. This method is generally more appropriate for squamous carcinomas of the esophagus [59]. The second method, (Vieth and Stolte system [51]), is more comprehensive and separates intraepithelial lesions from invasive carcinomas clearly. It categorizes M1–M4 levels, taking the duplication of the muscularis mucosae, which is frequently observed in Barrett’s esophagus, into consideration—M1: carcinoma limited to the lamina propria mucosae (no invasion of muscularis mucosae); M2: invasion into the superficial muscularis mucosae; M3: carcinoma infiltrating the layer between the superficial and deep muscularis mucosae, often referred to as the intervening layer or duplicated muscularis mucosae zone; M4: invasion into the deep muscularis mucosae. This system is therefore more appropriate for use in esophageal adenocarcinomas and requires larger specimens for the division between M3 and M4. The method used should be recorded in the report.

Submucosal invasion is measured from the deepest layer of the lamina muscularis mucosae to the deepest point of invasion. Generally, submucosal invasion is divided into three tiers (sm1—superficial one third of submucosa; sm2—intermediate one third of submucosa, and sm3—outer one third of submucosa). This division may be difficult, as it depends on the amount of submucosa included in the specimen (as in ER specimens) [60]. Since there is no muscularis propria for a landmark, the division is not accurate. For the classification of superficial neoplastic lesions measurements in microns has been recommended as an alternative [61] and should be considered standard as it provides a more accurate risk stratification for lymph node metastasis. In accordance with the increased risk for lymph node metastases for sm2 carcinomas, the threshold for sm2/sm3 has been defined as 500/1000 µm for adenocarcinomas and 200 µm for the sm2 category of squamous cell carcinomas.

### 8.4. Biomarker Testing

Biomarker testing may also be performed on ER tissue samples. However, depending on local or national guidelines, testing of early carcinomas may not be mandatory due to a lack of immediate clinical consequences, i.e., systemic treatment as administered for locally advanced or metastasized carcinomas. If upfront testing for therapeutically relevant biomarkers is performed, this should ideally include HER2, Mismatch Repair Deficiency /Microsatellite Instability Status, PD-L1, and Claudin18.2 for adenocarcinomas. For squamous cell carcinomas, only PD-L1 is currently used as a therapeutically relevant biomarker [62,63].

### 8.5. Pathology Reporting According to the International Collaboration on Cancer Reporting

Structured pathology reporting is increasingly recognized as essential for standardization, clarity, and interdisciplinary communication [55]. There exist reporting guidelines of various national pathology societies such as the CAP (College of American Pathologists (CAP) [64] or the Royal College of Pathologists (RCPath) [65]. The International Collaboration on Cancer Reporting (ICCR) aims at integrating pathological and clinical reporting and has developed evidence-based datasets for almost all tumor entities [18].

Besides datasets for esophagectomies [55] or gastrectomies, dedicated datasets for reporting local excisions of the esophagus and gastroesophageal junction [7] and stomach [17] have also been published. ICCR datasets distinguish between core (mandatory) and non-core (optional but recommended) reporting elements.

For ESD specimens from the esophagus or GEJ, core items include the anatomical site and tumor type (according to WHO classification), histological grade, maximal invasion depth with precise measurement in micrometers (particularly for submucosal invasion), resection margin status (lateral and deep), and the presence or absence of lymphovascular invasion. Tumor size in three dimensions, as well as presence of ulceration or Barrett-associated intestinal metaplasia, are also mandatory in Barrett-related neoplasia. Non-core items include background mucosal pathology (e.g., chronic esophagitis or intestinal metaplasia), degree of host inflammatory response, and presence of tumor budding. While not always required, inclusion of these features can provide additional context for clinical decision-making, especially in early adenocarcinomas of the GEJ.

For gastric cancers, core items comprise: endoscopic procedure (e.g., EMR, ESD, or other), tumor focality, tumor site (anatomical location), tumor dimensions (maximum size in mm), histological tumor type (WHO classification), histological grade (applicable only for tubular and papillary adenocarcinomas; well, moderate, poor, undifferentiated), layers present in specimen (lamina propria, muscularis mucosae, submucosa, etc.), extent of invasion (including depth of submucosal invasion in µm), absence or presence of lymphovascular invasion, and margin status (deep and lateral margins, including distance from closest margin in mm; also involvement of margins by high-grade or low grade dysplasia). Non-core items include clinical information (relevant biopsy results, prior cancer, risk conditions, indication for ER), macroscopic tumor type (e.g., Paris classification: 0-I, 0-IIa/b/c, 0-III), Laurén classification, additional tumor measurements (other than maximum dimension), specimen dimensions (overall, not just tumor), coexistent pathology (e.g., Helicobacter pylori gastritis, autoimmune gastritis, reactive gastropathy, intestinal metaplasia, polyps), previous neoadjuvant therapy details and response, ancillary studies that are not required for clinical management (e.g., neuroendocrine markers and Ki-67 for non-MiNEN/NEC), ancillary studies impacting management (e.g., HER2, MSI/MMR, EBV, as indicated for carcinoma type or context), and additional comments (e.g., tissue orientation issues, underestimation, recommendations for further management) [17,18].

## 9. Risk Factors and Their Clinical Significance

The risk of lymph node metastasis differs between organs and also between squamous cell carcinomas and adenocarcinomas [66,67,68,69]. For esophageal squamous cell carcinomas, oncological curative resection is defined for tumors showing the following features: pT1b (sm1) with an invasion depth less than 200 µm in the submucosal layer without risk factors (L0 and V0) and tumor differentiation of G1 or G2. For esophageal (Barrett’s) adenocarcinoma or adenocarcinomas of the esophagogastric junction, it is as follows: pT1b (sm1) with an infiltration depth up to 500 µm in the submucosal layer, L0 and V0, and tumor differentiation of G1 or G2 [70,71]. For gastric carcinomas ER is considered oncologically safe for differentiated non-ulcerated tumors < 2 cm diameter and L0 and V0 with up to a maximum of one of the expanded criteria: pT1b (sm1) with an infiltration depth up to 500 µm in the submucosal layer, a non-ulcerated lesion independent of size, a differentiated ulcerated lesion < 3 cm diameter, or an undifferentiated lesion < 2 cm diameter [72].

From a clinical perspective, the pathological findings following ESD determine whether the resection is curative. A complete (R0) resection without high-risk features supports continued endoscopic follow-up. In cases with high-risk histological features or incomplete vertical margin (R1), the multidisciplinary team may recommend additional surgical resection or adjuvant therapy. For incomplete lateral margins, follow-up biopsies for control and/or additional radiofrequency ablation may be sufficient and should be discussed in a multidisciplinary setting. After an R1 resection in dysplastic Barrett’s esophagus without invasive adenocarcinoma (i.e., positive margins after endoscopic submucosal dissection or mucosal resection), the optimal management strategy is determined by the risk profile of the residual neoplasia and patient factors [11,70,71] (Table 2).

Repeat endoscopy is recommended as first-line management for most patients with Barrett’s neoplasia limited to the mucosa (T1a) or superficial submucosa (sm1, <500 μm), provided there is no lymphovascular invasion and the lesion is well- or moderately differentiated. The American College of Gastroenterology and the American Gastroenterological Association both recommend that visible residual or recurrent lesions should undergo repeat endoscopic resection, while flat residual Barrett’s mucosa should be treated with ablation, most commonly radiofrequency ablation, to achieve complete eradication of intestinal metaplasia and reduce recurrence risk. Endoscopic therapy is preferred over esophagectomy for T1a and select low-risk T1b lesions, given comparable long-term survival and lower morbidity [54,56,59]

Esophagectomy is indicated if high-risk features are present—deep submucosal invasion (>500 μm), poor differentiation, or lymphovascular invasion—or if endoscopic therapy fails to achieve complete eradication after repeat attempts. Multidisciplinary evaluation is recommended in these cases.

Surveillance after endoscopic therapy should be performed at 3, 6, and 12 months, then annually, as suggested by the American Gastroenterological Association, to detect early recurrence [71]. All management decisions should incorporate shared decision-making, considering patient comorbidities and preferences [11,70,71].

The eCura score is a validated risk stratification system used to estimate the risk of lymph node metastasis (LNM) in patients with early gastric cancer (EGC) who have undergone non-curative endoscopic submucosal dissection (ESD) [73]. The score is calculated based on five pathological factors: lymphatic invasion (three points), tumor size > 30 mm (one point), positive vertical margin (one point), venous invasion (one point), and submucosal invasion ≥ 500 μm (one point). The total score ranges from 0 to 7, and patients are categorized into three risk groups: low- (0–1), intermediate- (2–4), and high- (5–7) risk for LNM [73]. This system helps guide post-ESD management decisions, such as whether to recommend additional surgery or observation. Multiple studies have demonstrated that the eCura score reliably predicts LNM risk and cancer-specific outcomes. High-risk patients have an approximately 25% rate of lymph node metastases and benefit most from additional surgery, while low-risk patients show lymph node metastases in only up to 3% and have excellent cancer-specific survival even without further intervention [74,75]. For patients in the low-risk eCura tier, endoscopic resection may therefore generally be considered sufficient, and management may consist of endoscopic and clinical surveillance without additional surgery, with a focus on regular follow-up to detect local recurrence or metachronous lesions. For patients in the intermediate-risk tier, management may be individualized, balancing the estimated risk of lymph node metastasis against patient age, comorbidities, and preferences. Additional gastrectomy with lymphadenectomy may be considered, but careful shared decision-making and multidisciplinary discussion are recommended. For patients in the high-risk eCura tier, additional gastrectomy with lymph node dissection may be recommended because of the substantial risk of lymph node metastasis. Observation alone may then generally be reserved for patients who are unfit for surgery or who decline further intervention after counseling. The eCura system is widely used in clinical practice in Japan and has been externally validated in large multicenter cohorts [76,77].

## 10. Pathologic Evaluation of Endoscopic Submucosal Dissection Versus Endoscopic Mucosal Resection Specimens

ESD specimens and EMR differ substantially in their histopathologic evaluability. The principal distinction arises from the method of resection: ESD typically achieves En bloc removal irrespective of lesion size, whereas EMR—particularly for lesions > 20 mm—frequently necessitates piecemeal excision [11,78]. En bloc resection in ESD preserves the anatomic relationships between the lesion, adjacent mucosa, and underlying submucosa, enabling precise assessment of both lateral and deep margins. Margin status can therefore be determined with high confidence, and the depth of submucosal invasion can be measured accurately. Furthermore, architectural preservation facilitates reliable identification of lymphovascular invasion (LVI) and characterization of histologic heterogeneity, including focal high-grade dysplasia or submucosal invasive carcinoma that may not be apparent endoscopically [41].

In contrast, the piecemeal nature of EMR inherently limits margin assessment, as fragmentation disrupts specimen orientation and spatial continuity. Depth of invasion may be underestimated or rendered indeterminate when submucosal tissue is incompletely sampled, and LVI detection is less reliable due to distortion from cautery and crush artifact. These limitations are particularly problematic for large, flat, or laterally spreading tumors, in which individual EMR fragments may fail to capture the deepest or most histologically aggressive areas [79,80].

Artifacts differ between the two modalities. EMR specimens more commonly exhibit crush artifact and coagulative distortion from snare excision, which can impair cytologic and stromal interpretation [11]. Particularly in piecemeal specimens, these artifacts may occur in any margin of any single specimen, causing major challenges for proper orientation, reconstruction, and subsequent determination of a final resection status. ESD specimens may also show localized thermal injury from electrosurgical knives, occasionally obscuring the epithelial–stromal interface at the margin. However, this is observed in a lower frequency and better to interpret when pre-intervention assessment could clearly show the demarcation of the target lesion that allows sufficient security distance to the margin of the specimen. EMR specimens may be generally simpler and faster to process in the pathology laboratory, but ESD specimens yield superior diagnostic details [16,59,80]. In general, ESD generates specimens that allow precise macroscopic work up, proper orientation, and further tissue processing for histology, allowing exact characterization of malignant and premalignant lesions and histological risk factors. Moreover, the precise determination of invasion depth in contrast to some EMR specimens makes ESD the preferred method for the oncologically safe removal of most early upper GI malignancies.

From a pathologist’s perspective, however, understanding the procedural steps and their impact on tissue morphology is essential for accurate evaluation, especially of margins, depth of invasion, and lymphovascular invasion (LVI). Although to a lesser amount compared to EMR, several steps during the procedure itself can cause challenges in the pathological evaluation:

For lesion marking the endoscopist first identifies the lesion and marks the perimeter approximately 5 mm beyond its visible edge using a coagulation device. These markings help guide resection boundaries. However, coagulation marks may appear histologically as small foci of cautery artifact and should not be mistaken for dysplasia or tumor extension [41].

For lifting the mucosa and creating a safety plane above the muscularis propria, a fluid (e.g., saline, glycerol, or hyaluronic acid, often with epinephrine or dye) is injected beneath the lesion. This submucosal lifting can create artificial tissue planes or spaces, which may mimic lymphatic channels. It is important not to misinterpret these artifacts as evidence of lymphovascular invasion. The injection may also lead to localized edema or stromal changes [16,80].

Mucosal incision is performed just outside the marked borders to allow access to the submucosa using an electrosurgical knife. Therefore, incision edges often show thermal injury, which can obscure margin assessment. Cautery artifacts at these points may mimic high-grade dysplasia or carcinoma in situ [41].

Finally, submucosal dissection continues in the submucosal plane using the same electrosurgical tools. Hemostasis is maintained using coagulation forceps or the knife itself. The goal is to remove the lesion in a single piece (En bloc) while preserving surrounding tissue. Thermal artifacts are therefore common, especially along the deep margin and at bleeding sites. Crush artifacts may occur due to manipulation with forceps, potentially distorting cellular architecture and complicating evaluation of dysplasia or invasion. In some cases, especially after prior biopsies, submucosal fibrosis is seen and can be misinterpreted as desmoplastic reaction or tumor-associated fibrosis [16].

## 11. Summary: Integrating Clinical and Pathological Perspectives in Endoscopic Resections for Upper GI Tumors

The integration of clinical and pathological perspectives is critical to maximizing the safety, efficacy, and long-term outcomes of endoscopic resection, particularly submucosal dissection in patients with upper gastrointestinal tumors (Table 3). This is important at different stages of diagnostics and therapy:

First, enhanced patient selection is achieved by clinical assessment through high-resolution endoscopy and imaging. This helps to identify which neoplastic lesions are best suited for ESD, focusing on those with superficial invasion and low risk of lymph node metastasis. The pathological evaluation of the diagnostic biopsy provides confirmation of malignancy, histologic type, and differentiation, ensuring that candidates for ESD meet evidence-based curability criteria [11,16,30,32,33,70].

Subsequently, accurate resection and histological assessment for risk stratification is a basis for tailored management. Here, ESD aims clinically for En-bloc resection, minimizing recurrence risk and preserving organ function. Pathologically, thorough examination of the resected specimen allows for confirmation and more detailed characterization of the tumor, including precise measurement of invasion depth, margin status, and detection of lymphovascular invasion. Additional biomarker testing can be performed if required. Additional findings, such as underlying intestinal metaplasia with or without dysplasia, inflammation, etc., should also be reported [8,11,16,32,33,70].

In summary, clinical–pathological correlation facilitates intra-procedural decisions, such as determining resection extent or halting the procedure if malignancy is more advanced than expected. This approach supports multidisciplinary collaboration, reduces adverse event rates, and aligns care with the most current evidence. Academically, integrating both perspectives supports the generation of robust data on ESD effectiveness and safety, which will offer clinicians and pathologists feedback loops that help refine criteria for ESD indications and post-procedure surveillance. This may particularly helpful in non-curative situations where the risk of surgery needs to be balanced against the risk of lymph node metastases.

## Figures and Tables

**Figure 1 jcm-14-08817-f001:**
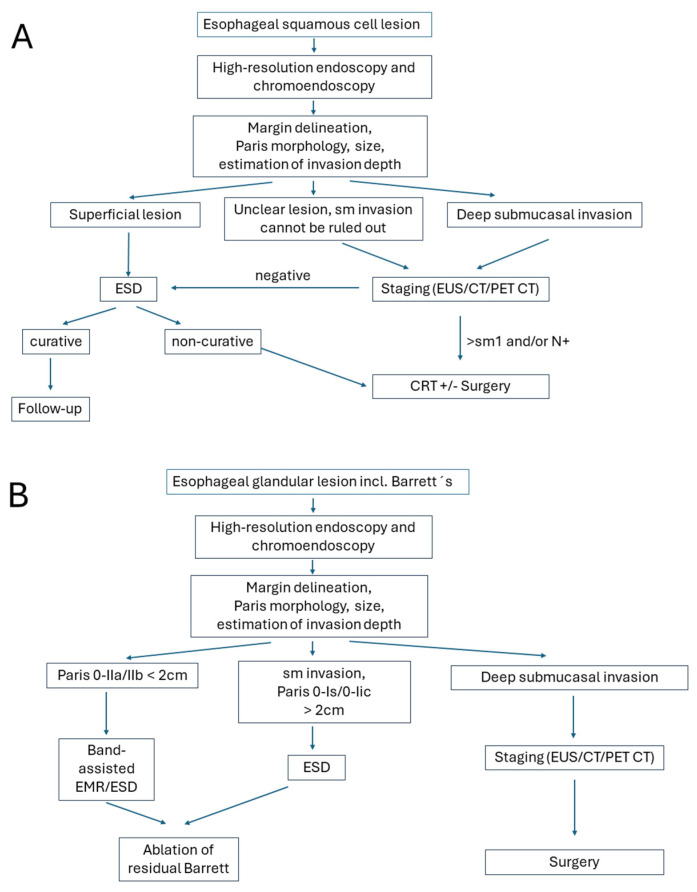
Algorithms for the treatment of early gastroesophageal cancers—endoscopic resection (ER) vs. surgery for (**A**) esophageal squamous cell lesion; (**B**) esophageal glandular lesion; (**C**) gastric lesion. Modified after [11].

**Figure 2 jcm-14-08817-f002:**
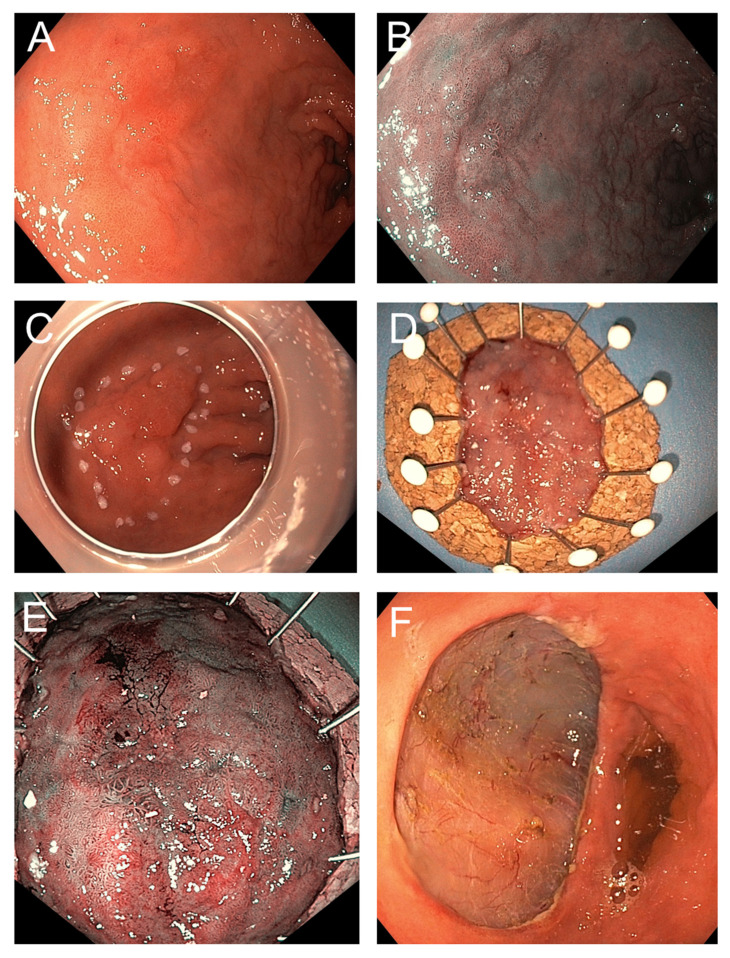
Endoscopic pre-procedure assessment and performance of ESD. (**A**) Endoscopical aspect of an irregular mucosal gastric lesion as seen with high-definition white-light endoscopy and (**B**) narrow-band imaging (NBI), suspicious for early gastric cancer. (**C**) In situ marking of the resection margins. (**D**) Lesion after excision under white light and (**E**) control under NBI. (**F**) Post resection site.

**Figure 3 jcm-14-08817-f003:**
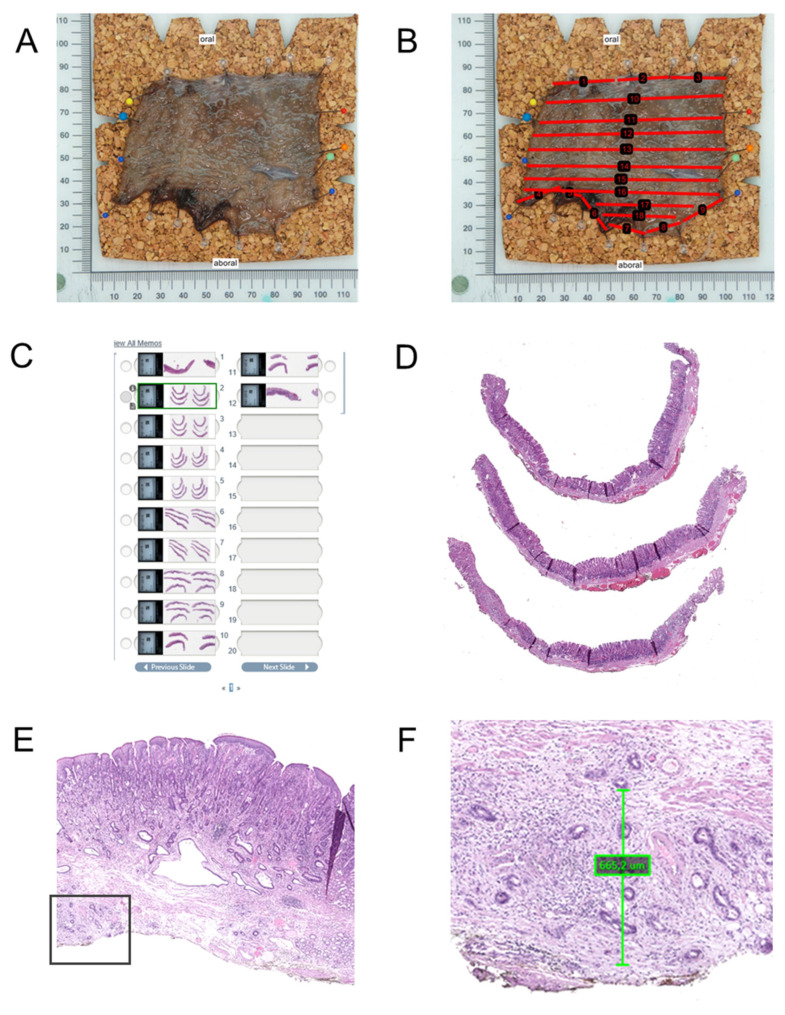
Aspects of macro- and histopathological work up of ESD specimens. (**A**) Pinned specimens as received from the endoscopy and after formalin fixation. (**B**) Schematic marks for sectioning. (**C**) Digital pathology tray with the complete case. (**D**) Medium-size magnification of a slide with three ESD sections. (**E**) Larger magnification of an esophageal adenocarcinoma with invasion of the submucosa. The box indicates the area of (**F**). (**F**) Digital measurement of submucosal invasion from the deepest layer of the (duplicated) lam. muscularis mucosae. Note a cancer gland at the inked vertical resection margin. This case has two high risk factors (>500 µm submucosal infiltration and positive vertical resection margin) for lymphatic spread and local recurrence.

**Figure 4 jcm-14-08817-f004:**
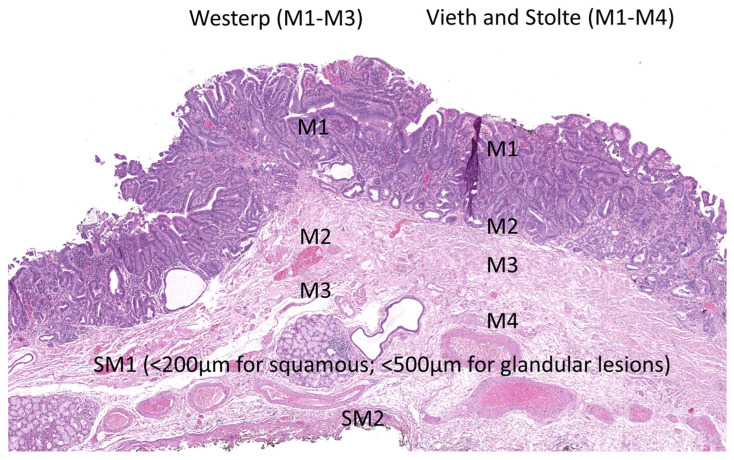
Subcategorization of mucosal and submucosal invasion of esophageal adenocarcinomas including the demonstration of the M1–M3 Westerp system and the M1–M4 Vieth and Stolte system for the classification of mucosal invasion.

**Table 1 jcm-14-08817-t001:** Vienna categories and corresponding histologic diagnosis and clinical consequences.

Vienna Category	Histologic Diagnosis	Clinical Consequence
1. Negative for neoplasia/ dysplasia	No neoplasia (benign, reactive changes)	Routine surveillance/ follow-up
2. Indefinite for neoplasia/ dysplasia	Uncertain, may be due to inflammation	Repeat biopsy or short-term follow-up
3. Low-grade neoplasia	Low-grade adenoma/ dysplasia	Endoscopic resection or close follow-up, depending on endoscopic features and risk factors
4. High-grade neoplasia	High-grade dysplasia, non-invasive carcinoma in situ, suspicious for carcinoma	Endoscopic resection (EMR/ESD) or surgery is strongly recommended
5. Invasive Neoplasia	Intramucosal carcinoma, submucosal invasion by carcinoma	Endoscopic resection (EMR/ESD) or surgery is strongly recommended. May require staging and MDT discussion

**Table 2 jcm-14-08817-t002:** Standard and extended risk factors for endoscopically resected upper gastrointestinal cancers.

Lesion	Standard	Extended
Esophagus-squamous	pT1a, m1, and m2 with no other risk factors for lymph node metastasis and radial resection margins.	pT1a, m1, m3, and pT1b sm1 (i.e., submucosal invasion ≤ 200 µm) with no other risk factors for lymph node metastasis and radial resection margin.
Esophagus-glandular (i.e., Barrett’s; also including carcinomas of the gastroesophageal junction)	pT1a with no histological risk factors for lymph node metastasis and completely resected.	pT1b sm1 (i.e., submucosal invasion ≤ 500 µm) with no other histological risk factors for lymph node metastasis and radial resection margin.
Stomach	pT1a, < 2 cm in diameter, with no other histological risk factors and with no ulceration.	No histological risk factors except as follows: 1. Size > 2 cm only; 2. Ulceration but < 3 cm; 3. Undifferentiated only; 4. <3 cm, pT1b (SM1, ≤500 µm) only.

**Table 3 jcm-14-08817-t003:** Summary of interdisciplinary clinical and pathological aspects.

Aspect	Clinical Perspective	Pathological Perspective	Enhanced Outcome
Patient Selection	Endoscopic staging	Histological confirmation	Accurate eligibility
Resection Strategy	En-bloc removal planning	Margin/vessel invasion status	Curative resection with low recurrence
Post-ESD Risk Assessment	Complication and relapse risk	Depth, type, lymphovascular invasion	Tailored follow-up/adjuvant therapy
Quality of Care	Multidisciplinary decision-making	Integrative diagnostic reporting	Improved patient-centric management

## Data Availability

No new data were created or analyzed in this study.

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
