# Peer review of "Endoscopic Submucosal Dissection (ESD) of Upper Gastrointestinal Carcinomas: An Integrated Clinical and Pathological Perspective"

_jcm, 2025, doi:10.3390/jcm14248817_

Round 1

Reviewer 1 Report

Comments and Suggestions for Authors

Comments: 

1. Although labeled “Review,” there is no Methods paragraph (databases, date range, key terms, inclusion/exclusion, approach to guideline synthesis). Please add a short, transparent Methods section and state the objective explicitly at the end of the Introduction. 

2. In several sections, guidelines, meta-analyses, registries, and single-center studies are cited together without distinguishing their evidence strength. It would help readers if you clearly indicated the study type either within the text or in a concise summary table. Adding simple tags such as “Guideline,” “Systematic review/meta-analysis,” “Prospective multicenter,” or “Retrospective single-center” would make it easier to gauge the reliability of each reference.

3. You already describe the endoscopic patterns and Paris classification clearly, but the section could be made more practical. Consider adding a compact checklist that helps assess resectability at the first inspection, including demarcation line clarity, microvascular irregularity, non-lifting after injection, and site-related difficulty. It would also be valuable to briefly outline common pitfalls, such as post-biopsy fibrosis that can mimic submucosal invasion or underestimation of depth in IIc and III lesions, and to link each to appropriate management decisions between ESD, EMR, or surgery. A single-page visual algorithm summarizing these points would make the guidance more accessible and clinically useful.

4. The section clearly explains the eCura system and its risk stratification, but it would benefit from brief practical guidance. Please add a short note on management by risk tier—for instance, observation for low-risk patients and gastrectomy for higher-risk cases.

5. Several long sentences could be simplified; consider professional editing for clarity. 

Author Response

Comment 1

Although labeled “Review,” there is no Methods paragraph (databases, date range, key terms, inclusion/exclusion, approach to guideline synthesis). Please add a short, transparent Methods section and state the objective explicitly at the end of the Introduction.

Reply 1

Thank you for this valuable remark. We have added a Methods section outlining the conceptual framework and approach of the paper. As indicated in the title—suggested by the editor of this special issue of the *Journal of Clinical Medicine*—the manuscript was not intended to be a systematic review, which represents a specific article category in the journal. Because the title already includes the term “Perspective,” we decided to maintain this designation to avoid any potential misunderstanding.

Comment2

In several sections, guidelines, meta-analyses, registries, and single-center studies are cited together without distinguishing their evidence strength. It would help readers if you clearly indicated the study type either within the text or in a concise summary table. Adding simple tags such as “Guideline,” “Systematic review/meta-analysis,” “Prospective multicenter,” or “Retrospective single-center” would make it easier to gauge the reliability of each reference.

Reply 2

Most of the references cited in this manuscript are review articles and guidelines, particularly those describing the ESD and EMR techniques and the pathological aspects. We chose to indicate this more generally in the Methods section rather than labeling each individual reference in the main text, in order to preserve readability. We hope this approach meets the reviewer’s expectations.

Comment 3

You already describe the endoscopic patterns and Paris classification clearly, but the section could be made more practical. Consider adding a compact checklist that helps assess resectability at the first inspection, including demarcation line clarity, microvascular irregularity, non-lifting after injection, and site-related difficulty. It would also be valuable to briefly outline common pitfalls, such as post-biopsy fibrosis that can mimic submucosal invasion or underestimation of depth in IIc and III lesions, and to link each to appropriate management decisions between ESD, EMR, or surgery. A single-page visual algorithm summarizing these points would make the guidance more accessible and clinically useful.

Reply 3

Thank you very much for this helpful recommendation. We have created a new figure that presents algorithms for three clinical scenarios (esophageal squamous cell lesions, esophageal glandular lesions, and gastric lesions). The challenging situations for endoscopic resection are already described mainly in the EMR section and partly in the ESD section.

In clinical practice, many of these situations are managed differently across centers (for example, the decision for EMR versus ESD), depending on the operator’s training and experience, the available equipment, and local treatment strategies. We believe that the manuscript already outlines the potential advantages of ESD over EMR, and that an exhaustive discussion with definitive recommendations would go beyond the scope of this ESD-focused perspective. Given that even current guidelines remain relatively vague on some of these points, we would kindly prefer to keep this aspect more open.

Comment 4

The section clearly explains the eCura system and its risk stratification, but it would benefit from brief practical guidance. Please add a short note on management by risk tier—for instance, observation for low-risk patients and gastrectomy for higher-risk cases.

Reply 4

We thank the reviewer for this helpful suggestion. In the revised manuscript, we have complemented the description of the eCura system with a brief note on typical management strategies by risk tier, indicating that low‑risk patients are usually managed with endoscopic follow‑up alone, whereas high‑risk patients are typically considered for additional gastrectomy with lymph node dissection. For the intermediate‑risk group, we now explicitly mention that management is individualized, taking into account patient comorbidities, preferences, and multidisciplinary discussion.”

Comment 5

Several long sentences could be simplified; consider professional editing for clarity.

Reply 5

Thank you very much for this hint. We carefully went through the manuscript and shortened longer sentences. Most parts of the text have already been checked by AI assisted language tools in the first version as stated in the acknowledgments. Before resubmission a final language check was performed.

Reviewer 2 Report

Comments and Suggestions for Authors

This is a good review article on endoscopic resection of upper GI malignancies.  I appreciate that the authors have organized it in an easy to read and follow format and have provided good visuals to support their manuscript.  The use of histopathology images that are clear and appropriately labeled along with the endoscopic images as well adds great value to the paper.  The differentiation between endoscopic and gastric lesions is also important which the authors have outlined.  The reference used are relevant and appropriate with many of them being from recent years.

Author Response

Comment

This is a good review article on endoscopic resection of upper GI malignancies.  I appreciate that the authors have organized it in an easy to read and follow format and have provided good visuals to support their manuscript.  The use of histopathology images that are clear and appropriately labeled along with the endoscopic images as well adds great value to the paper.  The differentiation between endoscopic and gastric lesions is also important which the authors have outlined.  The reference used are relevant and appropriate with many of them being from recent years.

Reply

We are very grateful for this encouraging and positive evaluation

Reviewer 3 Report

Comments and Suggestions for Authors

This is a well-structured comprehensive review on ESD for upper-GI carcinomas. The integrated clinical and pathological perspective provides a valuable resource for clinicians. The article is well-written and supported by high-quality, informative figures and tables.

I have the following comments:

  1. Consider adding a phrase to the Title clarifying the nature of this article (eg. a comprehensive/narrative review).

  2. Please add a brief Methods section describing the bibliographical databases searched and the keywords you used.
  3. In the "Biomarker Testing" section the authors state that testing "may not be mandatory." It could be helpful to briefly elaborate on why this varies (e.g., differences in regional guidelines, lack of consequence for early lesions) to provide more context for the reader.

  4. A proposed algorithm on the management of upper-GI malignancies based on your review and Table3 would significantly enhance the manuscript.

  5. Please check that all references are according to journal's format.

Author Response

Comment 1

Consider adding a phrase to the Title clarifying the nature of this article (eg. a comprehensive/narrative review).

Reply1

Thank you very much for your positive overall evaluation. This suggestion is similar to the comment made by Reviewer 1, and we therefore refer to our corresponding response. In brief, the manuscript is now labelled as a perspective, and a new Methods section has been added that more clearly describes its narrative structure.

Comment 2

Please add a brief Methods section describing the bibliographical databases searched and the keywords you used.

Reply 2

Please see our reply to comment 1

Comment 3

In the "Biomarker Testing" section the authors state that testing "may not be mandatory." It could be helpful to briefly elaborate on why this varies (e.g., differences in regional guidelines, lack of consequence for early lesions) to provide more context for the reader.

Reply 3

Thank you for pointing to this important issue. We re-wrote this chapter and hope that it is now better understandable.

Comment 4

A proposed algorithm on the management of upper-GI malignancies based on your review and Table3 would significantly enhance the manuscript.

Reply 4

Again, we refer to our reply to reviewer 1 who gave a similar suggestion. We created a new figure with algorithms for clinical decision making of the three entities (esophageal squamous cell lesions and glandular lesions and gastric lesions)

Comment 5

Please check that all references are according to journal's format.

Reply 5

We did a final check and completed the missing information of some references

Round 2

Reviewer 1 Report

Comments and Suggestions for Authors

Although labeled “Review,” there is no Methods paragraph (databases, date range, key terms, inclusion/exclusion, approach to guideline synthesis). Please add a short, transparent Methods section and state the objective explicitly at the end of the Introduction.